# CoLD: Co-evolutionary Latent Diffusion for MSA Generation

## Abstract

Protein structure prediction relies critically on Multiple Sequence Alignments (MSAs) that capture co-evolutionary information from homologous proteins. However, orphan proteins lacking sufficient homologs present a fundamental challenge, as sparse or absent MSAs severely limit folding accuracy. Current MSA generation methods operate through discrete token-based autoregressive generation, failing to capture the continuous nature of evolutionary relationships and global co-evolutionary constraints inherent in natural protein families. We introduce CoLD (Co-evolutionary Latent Diffusion), which reformulates MSA generation as conditional diffusion in the continuous embedding space of pretrained protein language models. By modeling evolution as smooth manifold trajectories and co-evolution through joint probability distributions over entire alignment embeddings, CoLD enables controllable homolog generation with biologically interpretable evolutionary distance control. Our two-stage training paradigm first establishes reliable embedding-to-sequence mappings, then optimizes diffusion with progressive biological constraints including profile consistency, sequence diversity, and amino acid distribution alignment. Extensive evaluation on CASP14/15 benchmarks and challenging zero-shot scenarios demonstrates that CoLD substantially outperforms existing methods, achieving above 11 point improvements in confidence metrics for orphan proteins while maintaining superior conservation pattern preservation (up to 0.994 correlation). These results validate the effectiveness of continuous diffusion modeling for capturing evolutionary relationships in protein sequence generation.

## 1 Introduction

Protein evolution unfolds through millions of years of natural selection, generating families of homologous sequences that preserve essential structural and functional elements while accumulating variation at non-critical positions. Multiple Sequence Alignments (MSAs) capture this evolutionary history by organizing these related sequences to reveal conserved domains, functional motifs, and co-evolutionary relationships that collectively encode the constraints governing protein folding and stability (Jumper et al., 2021). This evolutionary information has proven indispensable for modern structure prediction, with breakthrough methods like AlphaFold2 (Jumper et al., 2021) and RoseTTAFold (Baek et al., 2021) fundamentally dependent on extracting co-evolutionary signals from deep MSAs (Watson et al., 2023; Rao et al., 2021). However, when evolutionary data becomes sparse, prediction accuracy deteriorates dramatically. Orphan proteins, which lack detectable homologs and represent approximately 20% of metagenomic sequences (Chowdhury et al., 2022), experience substantial accuracy degradation compared to proteins with abundant evolutionary information (Yang & Zhang, 2023), highlighting the critical bottleneck imposed by MSA availability.

Traditional MSA construction relies on homology search algorithms such as HHblits (Remmert et al., 2012) and JackHMMER (Johnson et al., 2010), which are fundamentally constrained by database coverage and fail completely for orphan proteins. Recent advances in artificial intelligence have motivated a new generation of MSA generation methods (Zhang et al., 2023; Chen et al., 2024; Cao et al., 2025; Nijkamp et al., 2023) that synthesize virtual homologs to augment sparse alignments, demonstrating substantial improvements in downstream structure prediction tasks. However, these approaches predominantly employ autoregressive generation strategies and operate through sequential token prediction, resulting in error accumulation during long sequence generation (Bengio

et al., 2015) and fundamental difficulty in zero-shot scenarios where no existing MSA is available for conditioning. This gap between current methods and biological reality calls for fundamentally different approaches to modeling evolutionary sequence relationships.

As shown in the first panel of Figure 1, effective MSA generation requires modeling both the evolutionary process and co-evolutionary relationships through appropriate paradigms. Regarding **evolution**, protein language models such as ESM-2 (Lin et al., 2023) have learned continuous representations that capture smooth evolutionary distances and physicochemical constraints through pretraining on hundreds of millions of natural sequences. This learned manifold naturally encodes the gradual substitution relationships reflected in scoring matrices (Henikoff & Henikoff, 1992), making diffusion-based denoising an ideal paradigm for simulating the iterative refinement process underlying homologous sequence evolution. Regarding **co-evolution**, MSAs encode joint probability distributions over sequence ensembles where correlated mutations across positions maintain structural and functional constraints (Marks et al., 2011). The co-evolutionary signal emerges from statistical dependencies that span entire alignments, creating complex correlation patterns that define permissible evolutionary trajectories within protein families (Yang et al., 2024). Existing generative approaches that model sequences in-

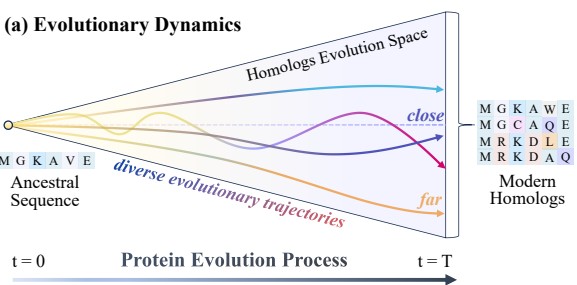

(a) Evolutionary Dynamics

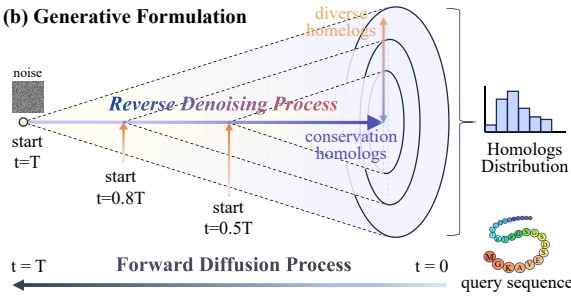

(b) Generative Formulation

Figure 1: **Evolutionary Modeling Through Diffusion.** **(a)** Protein evolution generates diverse homologs along continuous trajectories from ancestral sequences. **(b)** CoLD models this through controllable diffusion in embedding space, with variable starting timesteps $t_{\text{start}}$ controlling evolutionary divergence.

dependently fundamentally fail to capture these global correlations, limiting their ability to generate biologically coherent MSAs that preserve the intricate co-evolutionary relationships essential for accurate structure prediction. The biological intuition motivates to model both the evolutionary process and co-evolutionary relationships through diffusion-based paradigms

We introduce **CoLD** (Co-evolutionary Latent Diffusion), which models MSA generation as conditional diffusion in the continuous latent space of pretrained protein language models. CoLD captures evolutionary processes through iterative denoising that simulates gradual sequence divergence from ancestral states, while modeling co-evolutionary relationships by jointly generating entire MSA embeddings that preserve global statistical dependencies. Our framework incorporates ESM-guided attention mechanisms to capture residue conservation patterns and enables controllable generation of MSAs with varying evolutionary distances through noise scheduling at different diffusion timesteps.

Our main contributions are the following:

- **Evolutionary process simulation**: We pioneer diffusion-based modeling of protein evolution in continuous latent space, enabling smooth simulation of homologous sequence generation through progressive denoising that mirrors natural evolutionary processes.

- **Co-evolutionary joint distribution modeling**: CoLD models entire MSA embeddings jointly rather than generating sequences independently, capturing global co-evolutionary constraints through parallel diffusion that preserves alignment-level statistical structure.

- **Zero-shot capability with biological controls**: Our two-stage training paradigm with ESM-guided attention and controllable noise scheduling enables high-quality MSA generation from single queries while maintaining evolutionary coherence across generated ensembles.

## 2 RELATED WORK

**Multiple Sequence Alignment Generation.** The computational generation of MSAs has evolved from database-dependent retrieval methods to sophisticated AI-driven approaches that synthesize virtual homologous sequences. Early methods like HHblits (Remmert et al., 2012) and JackHM-MER (Johnson et al., 2010) perform exhaustive database searches but are inherently limited by sequence coverage and computational scalability. Contemporary generative approaches can be categorized into three paradigms: sequence inpainting methods such as MSA-Generator (Zhang et al., 2023) and EvoGen (Zhang et al., 2022) complete partial alignments using encoder-decoder architectures with axial attention; prompt-based conditional generation including MSAGPT (Chen et al., 2024) with 2D evolutionary positional encoding and EvoDiff-MSA (Alamdari et al., 2023) utilizing discrete diffusion on MSA prompts; and evolution space methods like PLAME (Cao et al., 2025) leveraging pretrained protein language model embeddings for direct generation. While demonstrating improvements, these methods predominantly employ autoregressive generation (Shin et al., 2021; Hawkins-Hooker et al., 2021; Repecka et al., 2021) or variational approaches (Riesselman et al., 2018; Sevgen et al., 2023; Li et al., 2023) operating on discrete sequences, failing to model the joint co-evolutionary distribution essential for zero-shot generation from single query sequences.

**Diffusion Models for Biological Sequence Generation.** Diffusion models have achieved remarkable success in protein structure generation, with RFdiffusion (Watson et al., 2023) pioneering backbone generation through fine-tuning RoseTTAFold on denoising tasks, alongside FoldingDiff (Wu et al., 2024) using angular representations, ProteinSGM (Lee & Kim, 2023) with score-based modeling, and FrameDiff (Yim et al., 2023) operating in SE(3) manifolds (Trippe et al., 2022; Bose et al., 2024). For sequence generation, EvoDiff (Alamdari et al., 2023) introduced discrete diffusion with order-agnostic autoregressive and D3PM corruption schemes trained on evolutionary-scale data, while recent advances include structure-sequence co-design (Lisanza et al., 2024; Campbell et al., 2024), multimodal diffusion approaches (Wang et al., 2025; Su et al., 2024), and taxonomic conditioning (Zhang et al., 2024). However, existing methods operate either in discrete sequence space or focus exclusively on structure generation, leaving unexplored the potential of continuous latent space diffusion for MSA-level generation that can capture smooth evolutionary relationships and global co-evolutionary dependencies simultaneously.

## 3 METHODOLOGY

### 3.1 PRELIMINARY

We formalize MSA generation as learning the conditional distribution $P(\mathcal{M}|q)$ where $\mathcal{M} = \{s_1, s_2, \ldots, s_M\}$ represents a multiple sequence alignment and $q \in \mathcal{A}^L$ denotes the query sequence over amino acid alphabet $\mathcal{A}$.

**Evolutionary Manifold Structure.** Let $\phi : \mathcal{A}^L \to \mathbb{R}^{L \times d}$ be a pretrained protein encoder that maps sequences to continuous embeddings. We postulate that protein evolution occurs within a smooth manifold $\mathcal{Z} \subset \mathbb{R}^{L \times d}$ where the geometric structure encodes evolutionary relationships. Formally, for any protein family, there exists a connected submanifold $\mathcal{M}_{\text{evo}} \subset \mathcal{Z}$ such that homologous sequences satisfy $\phi(s_i) \in \mathcal{M}_{\text{evo}}$ with geodesic distances reflecting evolutionary divergence.

The manifold exhibits two key properties: (1) *Local smoothness*: $\|\phi(s) - \phi(s')\|_2 \propto d_{\text{evo}}(s, s')$ where $d_{\text{evo}}$ denotes evolutionary distance, and (2) *Conservation structure*: functionally critical regions have lower local variance $\sigma^2(\phi(s)_{:,j}) \propto$ conservation score at position $j$.

**Joint Distribution Decomposition.** We decompose MSA generation into two coupled processes:

$$P(\mathcal{M}|q) = \int P(\mathbf{Z}|\phi(q)) \prod_{i=1}^{M} P(s_i|\mathbf{Z}_i) \, d\mathbf{Z}, \tag{1}$$

where $\mathbf{Z} = [\mathbf{Z}_1; \ldots; \mathbf{Z}_M] \in \mathbb{R}^{M \times L \times d}$ represents the MSA embedding tensor and $\mathbf{Z}_i = \phi(s_i)$.

The critical insight is modeling $P(\mathbf{Z}|\phi(q))$ as a joint distribution over the entire MSA tensor rather than factorizing over individual sequences. This preserves co-evolutionary constraints through statistical dependencies: for co-evolving positions $(j, k)$, the conditional independence structure satisfies:

$$\mathbf{Z}_{:,:,j} \not\perp \mathbf{Z}_{:,:,k} | \phi(q). \tag{2}$$

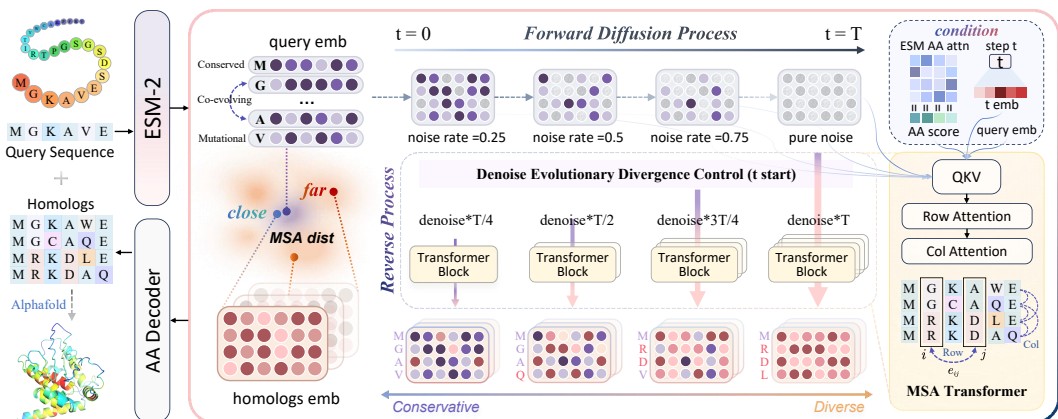

Figure 2: CoLD Architecture Overview. The framework employs ESM-2 encoder for continuous sequence representation, followed by diffusion-based MSA generation in embedding space. Controllable evolutionary divergence is achieved through variable starting timesteps $t_{\text{start}}$, with ESM-guided attention mechanisms preserving conservation patterns during denoising. The MSA Transformer processes joint embeddings through axial attention, capturing both intra-sequence dependencies and cross-sequence co-evolutionary relationships.

**Generation Paradigms.** The manifold structure enables continuous generation paradigms that respect evolutionary geometry. We parameterize $P(\mathbf{Z}|\phi(q))$ through a learned generative process $G_\theta : \mathbb{R}^{L \times d} \times \mathcal{N}(0, I) \to \mathbb{R}^{M \times L \times d}$ that maps query embeddings and random noise to MSA embeddings:

$$\mathbf{Z} \sim G_\theta(\phi(q), \epsilon), \quad \epsilon \sim \mathcal{N}(0, I^{M \times L \times d}). \tag{3}$$

The decoder $P(s_i|\mathbf{Z}_i)$ provides the mapping from continuous embeddings back to discrete sequences, completing the generation pipeline. This formulation naturally handles zero-shot scenarios where the query embedding $\phi(q)$ provides sufficient conditioning through its manifold position.

This formulation naturally handles zero-shot scenarios where the query embedding $\phi(q)$ provides sufficient conditioning through its manifold position. The theoretical optimality of this approach is established in Appendix A.

## 3.2 COLD ARCHITECTURE

Figure 2 presents the overall framework of our approach. CoLD employs ESM-2 (Lin et al., 2023) as the protein encoder $\phi$ to leverage evolutionary-scale pretraining for capturing sequence relationships in continuous space. This choice aligns with our evolutionary manifold hypothesis, as ESM-2's learned representations naturally encode physicochemical and evolutionary constraints through attention patterns trained on protein family data.

**MSA Representation and Encoding.** We represent the target MSA as a 3D tensor $\mathbf{Z} \in \mathbb{R}^{M \times L \times d}$ where each slice $\mathbf{Z}_{i,:,:} = \phi(s_i)$ corresponds to the embedding of sequence $s_i$. The query sequence is encoded as $\mathbf{z}_q = \phi(q) \in \mathbb{R}^{L \times d}$, serving as the primary conditioning signal. During generation, we initialize random noise $\mathbf{Z}_T \sim \mathcal{N}(0, \mathbf{I}^{M \times L \times d})$ and iteratively denoise to recover biologically coherent MSA embeddings.

**MSA Transformer Backbone.** The denoising network $\epsilon_\theta(\mathbf{Z}_t, \mathbf{z}_q, t, M)$ employs axial attention (Rao et al., 2021) to capture both intra-sequence dependencies and cross-sequence co-evolutionary patterns. For the MSA tensor $\mathbf{Z}_t \in \mathbb{R}^{M \times L \times d}$, row attention operates on $\mathbf{Z}_{t,i,:,:}$ to model positional relationships within sequences, while column attention on $\mathbf{Z}_{t,:,j,:}$ captures correlations across sequences at position $j$:

$$\text{Attn}_{\text{row}}(\mathbf{Z}_t)_{i,j,:} = \sum_{k=1}^{L} \alpha_{jk} \mathbf{Z}_{t,i,k,:}, \quad \text{Attn}_{\text{col}}(\mathbf{Z}_t)_{i,j,:} = \sum_{k=1}^{M} \beta_{ik} \mathbf{Z}_{t,k,j,:}. \tag{4}$$

We incorporate ESM-guided conservation attention by computing self-attention weights on the query embedding and using these to bias column attention toward conserved positions. The archi-

tecture includes adaptive layer normalization conditioned on timestep $t$ and target depth $M$ through learned scale and shift parameters.

**Amino Acid Decoder.** The decoder $D_\psi : \mathbb{R}^{L \times d} \to \mathbb{R}^{L \times |\mathcal{A}|}$ maps continuous embeddings to amino acid probability distributions.It consists of a multi-layer perceptron with GELU activation followed by softmax normalization, trained separately to establish reliable embedding-to-sequence mappings before joint diffusion training. The decoder enables controllable gap insertion through learned threshold mechanisms that determine when to predict gap tokens versus amino acids based on embedding magnitudes.

### 3.3 DIFFUSION PROCESS

We model MSA generation through conditional diffusion in the continuous embedding space, where the forward process simulates evolutionary divergence and the reverse process reconstructs coherent alignments from noise.

**Forward Process.** The forward diffusion process adds Gaussian noise to clean MSA embeddings $\mathbf{Z}_0$ according to a predefined variance schedule. At timestep $t$, the noisy MSA tensor follows:

$$q(\mathbf{Z}_t|\mathbf{Z}_0) = \mathcal{N}(\mathbf{Z}_t; \sqrt{\bar{\alpha}_t}\mathbf{Z}_0, (1 - \bar{\alpha}_t)\mathbf{I}), \tag{5}$$

where $\bar{\alpha}_t = \prod_{s=1}^t \alpha_s$ with $\alpha_t = 1 - \beta_t$ and $\beta_t$ following a cosine schedule. This schedule is biologically motivated: early timesteps preserve fine-grained evolutionary relationships while later timesteps approach pure noise, mirroring the natural process where recent evolutionary events are more detectable than ancient divergences.

**Reverse Process.** The reverse process generates MSA embeddings by iteratively denoising from random noise, conditioned on the query embedding $\mathbf{z}_q$:

$$p_\theta(\mathbf{Z}_{t-1}|\mathbf{Z}_t, \mathbf{z}_q) = \mathcal{N}(\mathbf{Z}_{t-1}; \boldsymbol{\mu}_\theta(\mathbf{Z}_t, \mathbf{z}_q, t), \sigma_t^2\mathbf{I}). \tag{6}$$

Following standard diffusion parameterization, we predict the noise $\boldsymbol{\epsilon}_\theta(\mathbf{Z}_t, \mathbf{z}_q, t)$ and compute the denoised prediction:

$$\boldsymbol{\mu}_\theta(\mathbf{Z}_t, \mathbf{z}_q, t) = \frac{1}{\sqrt{\alpha_t}}\left(\mathbf{Z}_t - \frac{\beta_t}{\sqrt{1 - \bar{\alpha}_t}}\boldsymbol{\epsilon}_\theta(\mathbf{Z}_t, \mathbf{z}_q, t)\right). \tag{7}$$

**Controllable Generation.** We enable controllable MSA generation by varying the starting timestep $t_{\text{start}} \leq T$ during inference. The key insight is that homologous sequences naturally cluster in the embedding manifold, making noise-corrupted query embeddings valid initialization points for MSA generation. We initialize the noisy MSA tensor as:

$$\mathbf{Z}_{t_{\text{start}}} \sim q(\mathbf{Z}_{t_{\text{start}}}|\mathbf{Z}_0^{\text{query}}), \tag{8}$$

where $\mathbf{Z}_0^{\text{query}} = \text{repeat}(\mathbf{z}_q, M) \in \mathbb{R}^{M \times L \times d}$ represents the query embedding replicated across $M$ sequences. This initialization leverages the biological principle that homologous sequences share common ancestry, with the noise level $t_{\text{start}}$ controlling evolutionary divergence time.

The generation process then applies the reverse diffusion from $t_{\text{start}}$ to 0:

$$\mathbf{Z}_0 = \mathcal{R}_\theta(\mathbf{Z}_{t_{\text{start}}}, \mathbf{z}_q; t_{\text{start}} \to 0), \tag{9}$$

where higher $t_{\text{start}}$ values induce greater sequence diversity while lower values preserve closer similarity to the query, providing interpretable control over the generated MSA's evolutionary span (Theorem A.4).

### 3.4 TRAINING STRATEGY

CoLD employs a two-stage training paradigm designed to decouple sequence validity from evolutionary modeling, followed by progressive constraint integration that mirrors biological hierarchy.

**Decoder Pretraining.** We first establish reliable continuous-to-discrete mappings by pretraining the decoder $D_\psi$ on natural protein sequences. This stage optimizes $\mathcal{L}_{\text{dec}} = -\mathbb{E}_{s \sim \mathcal{D}} \log P(s|\phi(s))$ where $\mathcal{D}$ represents the training distribution. Separating this objective prevents decoder instability from interfering with diffusion dynamics.

**Progressive Diffusion Training.** Diffusion training employs a composite loss function with time-dependent weighting to gradually introduce biological constraints:

$$\mathcal{L} = \mathcal{L}_{\text{diff}} + \lambda_{\text{profile}}(t)\mathcal{L}_{\text{profile}} + \lambda_{\text{div}}(t)\mathcal{L}_{\text{div}} + \lambda_{\text{kl}}(t)\mathcal{L}_{\text{kl}}. \tag{10}$$

The core diffusion loss follows standard formulation: $\mathcal{L}_{\text{diff}} = \mathbb{E}_{t,\epsilon} \|\epsilon - \epsilon_\theta(\mathbf{Z}_t, \mathbf{z}_q, t)\|^2$.

The profile consistency loss ensures the generated MSA's consensus matches the query structure: $\mathcal{L}_{\text{profile}} = \|\text{mean}(\mathbf{Z}_0) - \mathbf{z}_q\|^2$, which enforces that the MSA centroid aligns with the conditioning query embedding.The KL divergence loss aligns predicted amino acid distributions with evolutionary constraints: $\mathcal{L}_{\text{kl}} = \text{KL}(D_\psi(\mathbf{Z}_0)\|\mathbf{P}_{\text{target}})$, where $\mathbf{P}_{\text{target}}$ represents target amino acid frequencies derived from natural MSAs.The diversity regularization encourages sequence variation while preventing mode collapse: $\mathcal{L}_{\text{div}} = -\sum_{i<j} \|\mathbf{Z}_{0,i} - \mathbf{Z}_{0,j}\|^2$, promoting embedding diversity across generated sequences.

The progressive weighting schedule $\lambda(t)$ starts with pure diffusion training, then gradually introduces profile and KL constraints, and finally adds diversity regularization. This curriculum prevents conflicts between objectives and ensures stable convergence while maintaining both evolutionary plausibility and sequence diversity, with theoretical justification provided in Theorem A.3.

## 4 EXPERIMENT

### 4.1 EXPERIMENTAL SETUP

**Training Data.** We construct a comprehensive training dataset from PDB and UniClust30 datasets through systematic preprocessing. The PDB dataset contains 233,922 samples from 77,974 unique proteins with natural MSAs from three complementary databases: BFD/UniClust, MGnify, and UniRef90. The UniClust30 dataset provides 163,248 evolutionarily diverse sequences clustered at thirty percent identity. Our preprocessing enforces sequence length constraints up to one thousand residues and MSA depth requirements ranging from four to 128 homologs per family, yielding a final training set of 397,170 high-quality samples.

**Evaluation Datasets.** We assess CoLD performance across distinct evaluation scenarios. **CASP14** contains 61 solved protein structures while **CASP15** provides 64 additional targets, together comprising 125 gold-standard test cases from recent critical assessment competitions. **Zero-shot dataset** comprises 30 proteins of varying lengths absent from training data to evaluate true zero-shot generation capabilities.

**Baselines.** We compare against methods across two paradigms. **Single-sequence methods**: ESM-Fold (Lin et al., 2023) employs protein language model embeddings for direct structure prediction; AlphaFold2 (Jumper et al., 2021) operates without MSA information. **MSA-based methods**: AlphaFold2 with natural MSAs (Jumper et al., 2021), EvoDiff (Alamdari et al., 2023) using discrete diffusion, EvoGen (Zhang et al., 2022) with meta-generative frameworks, MSA-Generator (Zhang et al., 2023) utilizing sequence-to-sequence pretraining, MSAGPT (Chen et al., 2024) incorporating neural prompting, and PLAME (Cao et al., 2025) leveraging protein language model embeddings.

**Evaluation Metrics.** We employ four structural quality metrics: pLDDT measures per-residue confidence, LDDT evaluates local structural preservation, TM-score assesses global fold similarity, and GDT-TS quantifies overall structural accuracy across distance thresholds. All generated MSAs undergo AlphaFold2 structure prediction for fair comparison.

### 4.2 MAIN RESULTS

We evaluate CoLD against state-of-the-art baselines across three benchmarks to assess MSA generation quality through downstream structure prediction performance. Table 1 presents comprehensive results across all structural quality metrics.

The evaluation reveals distinct performance patterns across different scenarios. Zero-shot generation presents the most challenging setting, where CoLD achieves 62.03 pLDDT compared to 50.70 for PLAME, representing an 11.33 point improvement. This substantial gap indicates that continuous manifold modeling provides significant advantages when no evolutionary templates are available. On CASP benchmarks, performance improvements remain consistent but more modest, with 4-6 point gains across metrics, reflecting the inherent difficulty of these curated challenging targets.

The results demonstrate a clear methodological hierarchy. Single-sequence approaches achieve baseline performance around 40-45 pLDDT, while MSA-based methods substantially improve to 50-70 pLDDT range. Within MSA generation approaches, methods operating in continuous embedding space consistently outperform discrete sequence generation, which themselves exceed traditional database search augmentation. This progression supports the premise that protein evolu-

Table 1: Main Results on Structure Prediction Benchmarks. We report performance across CASP14, CASP15, and Zero Shot datasets using AlphaFold2 for structure prediction. Best results are **bolded**, second-best are underlined.

| Model | CASP14 | | | | CASP15 | | | | Zero Shot Dataset | | | |
|---|---|---|---|---|---|---|---|---|---|---|---|---|
| | pLDDT | LDDT | TM | GDT-TS | pLDDT | LDDT | TM | GDT-TS | pLDDT | LDDT | TM | GDT-TS |
| *Single Sequence Based* | | | | | | | | | | | | |
| ESMFold | 42.03 | 40.67 | 0.35 | 29.70 | 44.65 | 42.13 | 0.36 | 30.28 | 41.06 | 39.12 | 0.34 | 29.31 |
| AF2 | 42.71 | 39.31 | 0.31 | 28.25 | 42.90 | 39.70 | 0.32 | 28.44 | 40.19 | 38.92 | 0.31 | 28.54 |
| *MSA Based* | | | | | | | | | | | | |
| AF2 MSA | 52.61 | 50.38 | 0.42 | 42.90 | 41.25 | 51.64 | 0.43 | 40.38 | 40.19 | 38.92 | 0.31 | 28.54 |
| EvoDiff | 49.12 | 45.33 | 0.40 | 33.88 | 48.63 | 39.80 | 0.37 | 34.56 | 40.09 | 39.80 | 0.32 | 29.70 |
| EvoGen | 53.48 | 50.09 | 0.45 | 42.22 | 54.16 | 50.57 | 0.47 | 43.78 | 42.51 | 40.86 | 0.35 | 32.13 |
| MSAGen | 61.54 | 53.29 | 0.44 | 41.84 | 61.35 | 51.93 | 0.45 | 41.93 | 43.39 | 35.14 | 0.33 | 31.26 |
| MSAGPT | 65.42 | 54.36 | 0.51 | 44.94 | 64.35 | 56.10 | 0.58 | 46.40 | 45.91 | 42.51 | 0.34 | 33.78 |
| PLAME | 64.14 | 59.40 | 0.57 | 51.93 | 65.26 | 60.28 | 0.56 | 52.51 | 50.70 | 47.66 | 0.37 | 35.04 |
| **CoLD** | **68.43** | **63.48** | **0.60** | **56.98** | **69.79** | **62.32** | **0.61** | **57.46** | **62.03** | **55.91** | **0.51** | **45.82** |

Table 2: Ablation study demonstrating individual component contributions. Performance degradation with each removed component validates the necessity of CoLD's integrated design.

| Method | CASP14 | | | | CASP15 | | | |
|---|---|---|---|---|---|---|---|---|
| | pLDDT | LDDT | TM | GDT-TS | pLDDT | LDDT | TM | GDT-TS |
| **CoLD (Full)** | **68.43** | 63.48 | **0.60** | **56.98** | 69.79 | **62.32** | **0.61** | **57.46** |
| w/o ESM Guidance | 65.72 | 61.15 | 0.55 | 52.34 | **70.12** | 58.94 | 0.57 | 53.81 |
| w/o Profile Loss | 64.38 | **64.21** | 0.53 | 50.76 | 66.45 | 59.73 | 0.56 | 52.18 |
| w/o Diversity Loss | 67.91 | 63.82 | 0.58 | 54.29 | 69.83 | 60.47 | 0.59 | 55.63 |
| w/o KL Loss | 66.24 | 60.19 | 0.57 | 55.43 | 68.17 | 61.58 | 0.58 | 54.92 |
| w/o Progressive Training | 62.85 | 56.74 | 0.51 | 48.67 | 64.29 | 55.36 | 0.52 | 49.75 |

tionary relationships are more effectively captured through learned continuous representations than discrete token manipulation.

## 4.3 ABLATION STUDY

We conduct systematic ablation experiments to validate each component's contribution in CoLD's architecture. Table 2 reveals nuanced interactions between architectural components, with some ablations occasionally outperforming the full model on specific metrics.

Progressive training proves most critical overall, with its removal causing consistent degradation across both datasets. Interestingly, removing diversity loss yields competitive performance and even achieves second-best results on several metrics, suggesting potential hyperparameter optimization opportunities in the full model. The w/o ESM Guidance variant achieves the highest pLDDT on CASP15, indicating that conservation guidance may occasionally over-constrain generation for certain protein families. Profile loss removal shows mixed effects, achieving best LDDT performance on CASP14 while degrading other metrics, highlighting the complex trade-offs between different structural quality measures. These results demonstrate that CoLD's components interact in sophisticated ways, with optimal configurations potentially varying across different evaluation contexts and protein characteristics.

## 4.4 EVOLUTIONARY DISTANCE CONTROL

We investigate the relationship between diffusion timesteps and evolutionary distance by systematically varying the starting ratio during generation. Table 3 presents structural quality metrics across different timestep ratios, while Figure 4 demonstrates the visual progression of MSA characteristics under controllable evolutionary divergence.

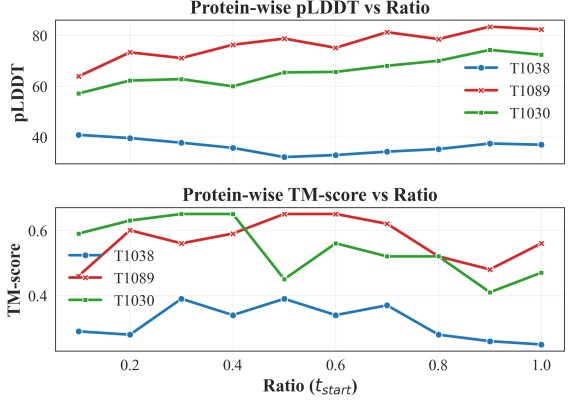

| Ratio | Similarity | pLDDT | TM |
|-------|-----------|-------|------|
| 10% | 6.80% | 55.03 | 0.61 |
| 20% | 10.81% | 61.23 | 0.65 |
| 30% | 17.83% | 63.16 | 0.66 |
| 40% | 30.24% | 65.74 | **0.67** |
| 50% | 47.33% | 64.70 | 0.56 |
| 60% | 67.83% | 67.65 | 0.58 |
| 70% | 84.36% | 67.82 | 0.53 |
| 80% | 92.63% | 69.99 | 0.53 |
| 90% | 94.00% | 70.96 | 0.42 |
| 100% | 94.13% | **71.88** | 0.48 |

Figure 3: Protein-specific evolutionary distance optimization showing target-dependent performance patterns across different noise ratios.

Table 3: Evolutionary distance control reveals divergent optima for different structural metrics across timestep ratios.

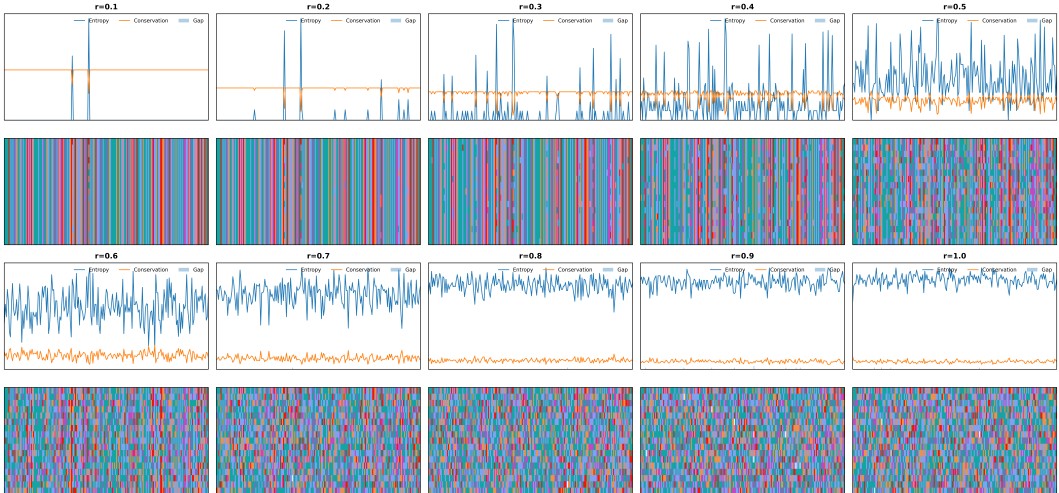

Figure 4: **Controllable Evolutionary Divergence in MSA Generation.** CoLD enables precise control over evolutionary distance through variable starting timestep ratios ($r = t_{\text{start}}/T$). From left to right (r=0.1 to r=1.0), generated MSAs exhibit increasing sequence diversity while preserving conservation patterns. Upper panels show statistical profiles (entropy, conservation, gap rate) and lower panels display sequence alignments. This controllability is unique to CoLD's continuous manifold approach and enables biologically interpretable MSA generation.

Figure 4 reveals three distinct evolutionary regimes across the controllable generation spectrum. At low ratios (r=0.1-0.3), generated MSAs exhibit minimal diversity with near-uniform conservation patterns, effectively producing close homologs with subtle variations that preserve structural integrity. The intermediate regime (r=0.4-0.7) demonstrates balanced evolutionary sampling where entropy increases selectively at variable positions while maintaining conservation at functionally critical sites—this regime corresponds to the optimal TM-Score performance observed in Table 3. At high ratios (r=0.8-1.0), the method generates distantly related sequences with maximum diversity, producing MSAs that capture broad evolutionary space but sacrifice local structural constraints for global sequence coverage.

Sequence similarity analysis confirms that timestep ratios directly control evolutionary divergence, with similarity decreasing monotonically from 94.13% at 100% ratio to 6.80% at 10% ratio. However, structural metrics exhibit divergent optimization patterns: pLDDT peaks at maximum divergence (100% ratio) while TM-Score optimizes at intermediate levels (40% ratio). This divergence reflects distinct information requirements where confidence metrics benefit from diverse evolutionary sampling while structural accuracy requires balanced homolog representation.

The conservation profile analysis in Figure 4 reveals that CoLD maintains position-specific conservation patterns across all generation ratios, unlike sequence-independent methods that treat all

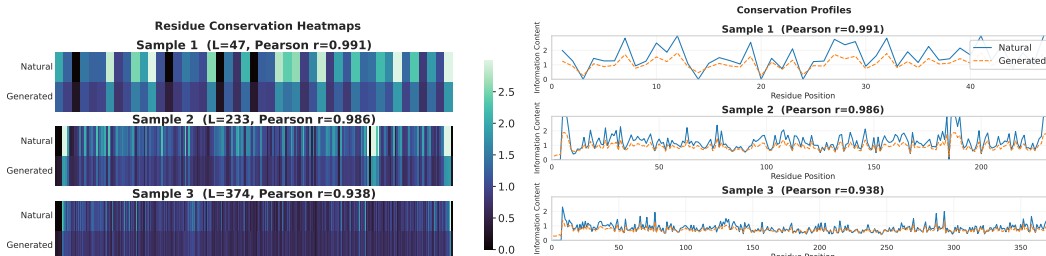

(a) Conservation heatmaps reveal spatial consistency between natural and generated MSAs across diverse protein lengths.

(b) Position-wise conservation profiles demonstrate quantitative preservation of functional constraints with high correlation.

Figure 5: Conservation analysis demonstrates accurate preservation of evolutionary constraints. Generated MSAs achieve Pearson correlations of 0.949-0.994 with natural conservation patterns across proteins of varying structural complexity.

positions uniformly. Notably, functionally important regions (evident as conservation peaks) remain preserved even at high diversity ratios, indicating that the continuous manifold embedding captures and maintains structural constraints inherent in the query protein's evolutionary neighborhood. This biological fidelity, combined with controllable diversity, enables adaptive MSA generation strategies tailored to specific downstream applications.

### 4.5 CONSERVATION PATTERN ANALYSIS

We evaluate conservation preservation by comparing information content profiles between generated and natural MSAs across representative protein targets. Conservation is quantified using $I = \log(20) - H$, where $H = -\sum_a f_a \log f_a$ represents Shannon entropy over amino acid frequencies at each position.

Figure 5 demonstrates strong conservation pattern preservation across diverse protein families. The heatmap visualization in Figure 5a reveals spatial conservation consistency, with functionally critical regions maintaining similar information content distributions between natural and generated alignments. This global correspondence indicates successful preservation of large-scale evolutionary constraints that govern protein family structure.

Position-wise analysis in Figure 5b provides quantitative validation with generated MSAs achieving Pearson correlations of 0.949-0.994 across targets of varying lengths and complexities. The profiles demonstrate accurate reproduction of both conserved structural elements and variable regions, capturing the detailed conservation landscapes that characterize authentic evolutionary relationships. Notably, correlation strength remains consistently high regardless of protein length, from compact 47-residue domains to extended 374-residue structures.

This conservation preservation stems from joint MSA embedding modeling, which maintains statistical dependencies between positions encoding co-evolutionary relationships. Unlike independent sequence generation, our approach captures correlation structures underlying functional constraints by processing entire alignment representations simultaneously in continuous space, enabling the preservation of evolutionary information essential for biological authenticity and structure prediction accuracy.

## 5 CONCLUSION

We present CoLD, a diffusion-based approach that models MSA generation in continuous protein embedding manifolds rather than discrete sequence space. This paradigm shift addresses fundamental limitations of autoregressive methods by capturing co-evolutionary relationships through joint distribution modeling and enabling zero-shot generation from single queries. Experimental results demonstrate substantial improvements over existing approaches, particularly in data-scarce scenarios where evolutionary information is limited. The controllable generation mechanism provides principled evolutionary distance manipulation, while conservation analysis validates biological authenticity. These contributions establish continuous manifold diffusion as an effective framework for MSA generation, advancing computational approaches to protein structure prediction and evolutionary analysis.

ETHICS STATEMENT

This work introduces COLD, a computational method that generates multiple sequence alignments (MSAs) to support protein structure prediction. Our study uses only publicly available protein sequence resources and does not involve human subjects, personal data, or clinical interventions. No wet-lab experiments were conducted, and we neither infer nor evaluate pathogenicity or toxicity. While improvements in protein modeling can have dual-use implications, our contribution is limited to MSA generation for scientific understanding of evolutionary constraints and for improving structure prediction on proteins with scarce homologs. We adhere to the licenses and terms of use of the underlying databases and cite all sources. To promote responsible research, we document our methods transparently, refrain from releasing any functionality intended for sequence design or activity optimization, and will distribute code and pretrained weights for non-commercial research under a terms-of-use agreement upon acceptance.

REPRODUCIBILITY STATEMENT

We aim to make COLD fully reproducible. The appendix lists all hyperparameters, model configurations, optimizer settings, learning-rate schedules, and loss weights used in every experiment. We provide exact data preprocessing scripts to reconstruct the training corpus from public snapshots of PDB-derived MSAs and UniClust30, including sequence length filters and MSA-depth ranges. Evaluation follows standard CASP14/15 targets and metrics, with AlphaFold2 inference settings (model variant, recycles, ranking metric) specified for comparability. We fix and report random seeds, log software versions (Python/PyTorch/CUDA), and note representative hardware (GPUs and memory) used. To facilitate replication, we will release: (i) complete source code with experiment configs, (ii) training and evaluation scripts, (iii) checkpoints and data manifests (with checksums), and (iv) instructions to reproduce all tables and figures from raw logs. Where nondeterminism may arise (e.g., CUDA kernels), we include deterministic flags and document any residual variance across runs.

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

# A  THEORETICAL ANALYSIS

## A.1  CO-EVOLUTIONARY STRUCTURE PRESERVATION

> **Proposition A.1**
>
> *Diffusion processes preserve co-evolutionary correlations in embedding space.*

Let $C_{ij} = \text{MI}(\mathbf{Z}_{:,i,:}, \mathbf{Z}_{:,j,:})$ denote mutual information between positions $i, j$ in MSA embedding tensor $\mathbf{Z} \in \mathbb{R}^{M \times L \times d}$. The forward diffusion process:

$$q(\mathbf{Z}_t | \mathbf{Z}_0) = \mathcal{N}(\mathbf{Z}_t; \sqrt{\bar{\alpha}_t} \mathbf{Z}_0, (1 - \bar{\alpha}_t)\mathbf{I}), \tag{11}$$

preserves co-evolutionary structure when noise addition respects correlation patterns encoded in the embedding space.

**Analysis:** The correlation coefficient between positions after $t$ diffusion steps becomes:

$$\rho_t(i, j) = \sqrt{\bar{\alpha}_t}\rho_0(i, j) + \sqrt{1 - \bar{\alpha}_t}\epsilon_{ij}, \tag{12}$$

where $\epsilon_{ij}$ represents noise correlation. Since ESM embeddings encode co-evolutionary relationships through attention mechanisms, the correlation structure $\rho_0(i, j)$ reflects biological dependencies that are preserved during the diffusion process.

## A.2  MANIFOLD CONVERGENCE PROPERTIES

> **Analysis A.2**
>
> *The reverse diffusion process converges to the evolutionary manifold.*

Consider the protein embedding manifold $\mathcal{M}_{\text{evo}} \subset \mathbb{R}^{L \times d}$ where evolutionarily related sequences concentrate. The reverse diffusion follows:

$$p_\theta(\mathbf{Z}_{t-1} | \mathbf{Z}_t, \mathbf{z}_q) = \mathcal{N}(\boldsymbol{\mu}_\theta(\mathbf{Z}_t, \mathbf{z}_q, t), \sigma_t^2 \mathbf{I}), \tag{13}$$

where $\boldsymbol{\mu}_\theta$ approximates the score function $\nabla_{\mathbf{z}_t} \log p(\mathbf{Z}_t | \mathbf{z}_q)$.

**Convergence Argument:** By construction of ESM pretraining on evolutionary data, the conditional distribution $p(\mathbf{Z} | \mathbf{z}_q)$ naturally concentrates on $\mathcal{M}_{\text{evo}}$. The reverse SDE converges to this manifold as the learned score function guides samples toward regions of high evolutionary probability. Query conditioning $\mathbf{z}_q$ ensures generated sequences remain within the appropriate protein family submanifold.

## A.3  INFORMATION-THEORETIC OPTIMALITY

> **Claim A.3**
>
> *Joint MSA modeling maximizes evolutionary information retention.*

Define total correlation $TC(\mathcal{M}) = \sum_i H(\mathbf{Z}_i) - H(\mathbf{Z})$ measuring statistical dependencies across MSA sequences. Joint modeling achieves:

$$TC_{\text{joint}} = \int P(\mathbf{Z} | \mathbf{z}_q) \log \frac{P(\mathbf{Z} | \mathbf{z}_q)}{\prod_i P(\mathbf{Z}_i | \mathbf{z}_q)} d\mathbf{Z}, \tag{14}$$

while independent sequence generation gives $TC_{\text{independent}} = 0$ by definition.

**Information Retention:** By the data processing inequality and properties of mutual information:

$$D_{KL}(P_{\text{natural}}(\mathbf{Z} | \mathbf{z}_q) \| P_{\text{joint}}(\mathbf{Z} | \mathbf{z}_q)) \leq D_{KL}(P_{\text{natural}}(\mathbf{Z} | \mathbf{z}_q) \| P_{\text{independent}}(\mathbf{Z} | \mathbf{z}_q)). \tag{15}$$

Joint modeling thus achieves lower approximation error to the natural MSA distribution and preserves more co-evolutionary information essential for structure prediction.

## A.4 EVOLUTIONARY DISTANCE CONTROL

> **Proposition A.4**
>
> *Starting timestep $t_{start}$ provides monotonic control over evolutionary distance.*

For generation initialized at timestep $t_{\text{start}}$, the expected evolutionary distance satisfies:

$$\mathbb{E}[d_{\text{evo}}(\text{generated}, \text{query})] = f\left(\frac{1 - \bar{\alpha}_{t_{\text{start}}}}{\bar{\alpha}_{t_{\text{start}}}}\right), \tag{16}$$

where $f$ is monotonically increasing and $d_{\text{evo}}$ measures sequence-level evolutionary divergence.

**Controllability Analysis:** The initialization $\mathbf{Z}_{t_{\text{start}}} \sim \mathcal{N}(\text{repeat}(\mathbf{z}_q, M), \frac{1 - \bar{\alpha}_{t_{\text{start}}}}{\bar{\alpha}_{t_{\text{start}}}} \mathbf{I})$ determines the noise level around the query embedding. Since embedding distance in the learned ESM space correlates with evolutionary distance, and $\bar{\alpha}_t$ decreases monotonically with $t$, higher $t_{\text{start}}$ values yield proportionally larger evolutionary distances. This provides interpretable biological control over the generation process.

# B IMPLEMENTATION DETAILS

## B.1 TRAINING INFRASTRUCTURE AND CONFIGURATION

We implement CoLD using PyTorch with distributed data parallel (DDP) training across 4 NVIDIA A100 GPUs. The training process spans approximately 500,000 optimization steps over 72 hours, employing mixed-precision training with automatic gradient scaling for computational efficiency. Our implementation utilizes gradient accumulation with a step size of 32 to simulate larger effective batch sizes while maintaining memory constraints.

## B.2 MODEL ARCHITECTURE SPECIFICATIONS

Table 4 presents comprehensive hyperparameter configurations for our CoLD framework. The architecture employs ESM-2 (t12_35M_UR50D) as the frozen protein encoder, providing 480-dimensional embeddings that capture evolutionary relationships learned from 65M protein sequences. The diffusion backbone consists of 4 transformer blocks with 512 hidden dimensions and 8 attention heads, processing MSA tensors up to $16 \times 1000 \times 480$ dimensions corresponding to maximum depth, sequence length, and embedding dimensionality.

## B.3 TRAINING PROTOCOL AND PROGRESSIVE OPTIMIZATION

Our two-stage training paradigm begins with decoder pretraining for 2 epochs using cross-entropy loss to establish reliable embedding-to-sequence mappings. The decoder architecture employs a 2-layer MLP with GELU activation, mapping 480-dimensional ESM embeddings to 21-dimensional amino acid distributions including gap tokens.

Diffusion training follows a progressive loss scheduling strategy implemented through training step ratios. Early training (steps 0-50%) employs pure diffusion loss $\mathcal{L}_{\text{diff}}$ to establish basic denoising capabilities. Middle training (50%-80%) introduces profile consistency loss with linearly increasing weight, ensuring generated MSA centroids align with query embeddings. Late training (80%-100%) incorporates full composite loss including diversity regularization and KL divergence alignment with natural amino acid distributions.

## B.4 DATA PROCESSING AND AUGMENTATION

We construct training datasets from PDB (233,922 samples) and UniClust30 (163,248 samples), applying systematic preprocessing including sequence length filtering (no more than 1000 residues) and MSA depth requirements (4-128 homologs). Encoded homologs undergo token mapping from 33-dimensional ESM vocabulary to our 21-dimensional amino acid space, with special handling for gap token insertion based on learned probability thresholds.

Table 4: Implementation Details and Hyperparameters

| Model Architecture | | Training Configuration | |
|---|---|---|---|
| ESM Encoder | ESM2-t12-35M | Batch Size | 1 (per GPU) |
| ESM Embedding Dim | 480 | Grad Accumulation | 32 steps |
| Hidden Dimension | 512 | Effective Batch Size | 128 |
| Transformer Layers | 4 | Mixed Precision | FP16 |
| Attention Heads | 8 | Max Grad Norm | 1.0 |
| Max Sequence Length | 1000 | Weight Decay | 0.01 |
| Max MSA Depth | 16 | Pin Memory | True |
| Vocab Size | 21 | Num Workers | 4 |
| Diffusion Parameters | | Loss Weights | |
| Timesteps $T$ | 1000 | Profile Weight $\lambda_p$ | 0.002 |
| Beta Schedule | Cosine | Diversity Weight $\lambda_d$ | 0.005 |
| Beta Start | $1 \times 10^{-4}$ | KL Weight $\lambda_{kl}$ | $5 \times 10^{-5}$ |
| Beta End | 0.02 | Gap Threshold | 0.08 |
| Optimization | | Computational Resources | |
| Decoder LR | $3 \times 10^{-4}$ | GPUs | $4 \times$ A100 |
| Diffusion LR | $1 \times 10^{-4}$ | Total Steps | 500k |
| LR Schedule | Cosine Annealing | Training Time | 72 hours |
| Optimizer | AdamW | Memory per GPU | 78GB |

Dynamic MSA depth sampling during training randomly selects target depths from [4, 8, 16] to promote generalization across varying alignment sizes. Amino acid distributions are normalized using softmax after concatenating rare amino acid probabilities into gap categories, ensuring valid probability distributions for KL divergence computation.

### B.5 COMPUTATIONAL OPTIMIZATION AND MEMORY MANAGEMENT

Memory efficiency is achieved through several optimizations: (1) embedding caching with LRU policy limiting cache size to 1000 entries, (2) gradient checkpointing for transformer blocks, (3) periodic GPU memory clearing every 50 inference steps during generation, and (4) distributed training with synchronized batch normalization across GPUs.

The cosine noise scheduler employs $s = 0.008$ offset parameter for improved training stability, while adaptive layer normalization conditions on both timestep and MSA depth embeddings through sinusoidal encoding. Generation employs DDIM sampling with $\eta = 0.0$ for deterministic inference, enabling controllable evolutionary distance through starting timestep manipulation.

## C STRUCTURAL PREDICTION CASE STUDIES

We present representative structure prediction results demonstrating the downstream impact of CoLD-generated MSAs across diverse protein targets. Each case compares AlphaFold2 predictions using MMseqs2-sampled alignments as baseline against CoLD-generated MSAs, both providing 16 homologous sequences for structure prediction.

The structural comparisons across Figures 6-11 reveal three distinct performance patterns across protein families. T1082 demonstrates CoLD's most significant advantage, with substantial improvements in both fold accuracy (TM-score: 0.815 vs 0.548) and confidence (pLDDT: 66.9 vs 45.0), exemplifying scenarios where continuous manifold sampling captures evolutionary relationships missed by discrete sequence retrieval. T1056 shows improved confidence (pLDDT: 43.4 vs 31.7) with comparable structural accuracy, indicating enhanced prediction reliability in challenging low-

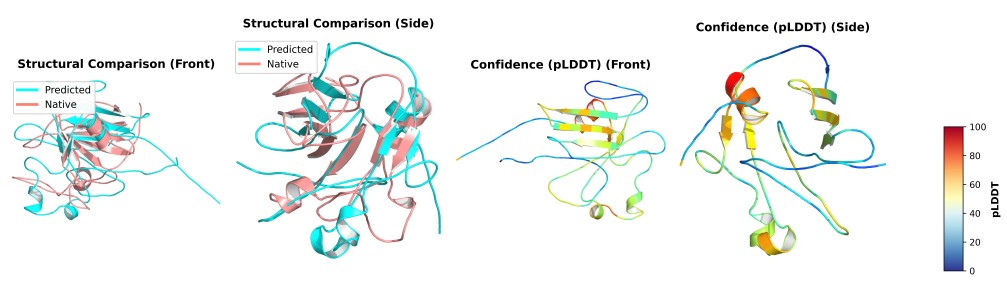

**T1056: pLDDT=31.7 | pTM=0.280 | TM-score=0.352 | MMseqs2 MSA**

Figure 6: T1056 baseline using MMseqs2 MSA: pLDDT=31.7, TM-score=0.352

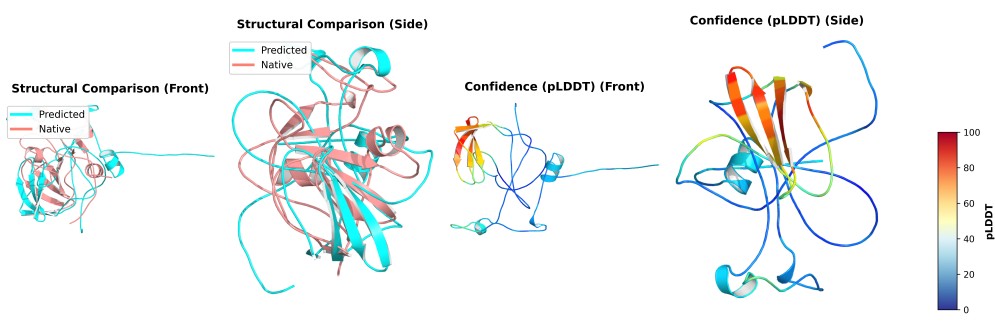

**T1056: pLDDT=43.4 | pTM=0.440 | TM-score=0.364 | CoLD MSA**

Figure 7: T1056 using CoLD-generated MSA: pLDDT=43.4, TM-score=0.364

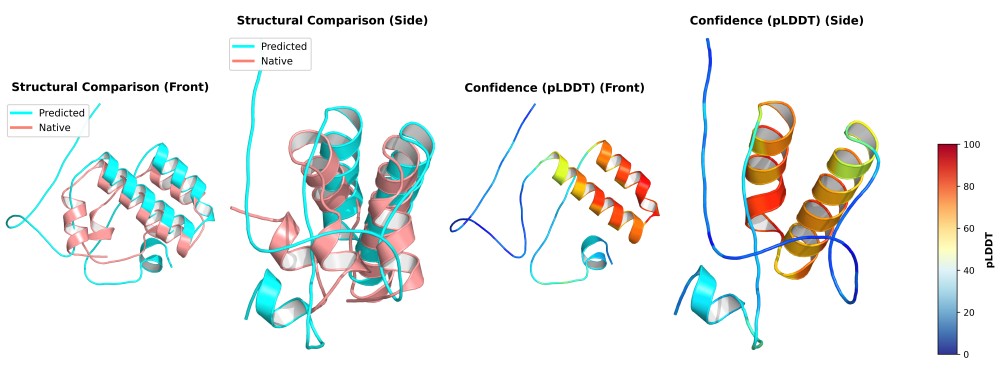

**T1082: pLDDT=45.0 | pTM=0.400 | TM-score=0.548 | MMseqs2 MSA**

Figure 8: T1082 baseline using MMseqs2 MSA: pLDDT=45.0, TM-score=0.548

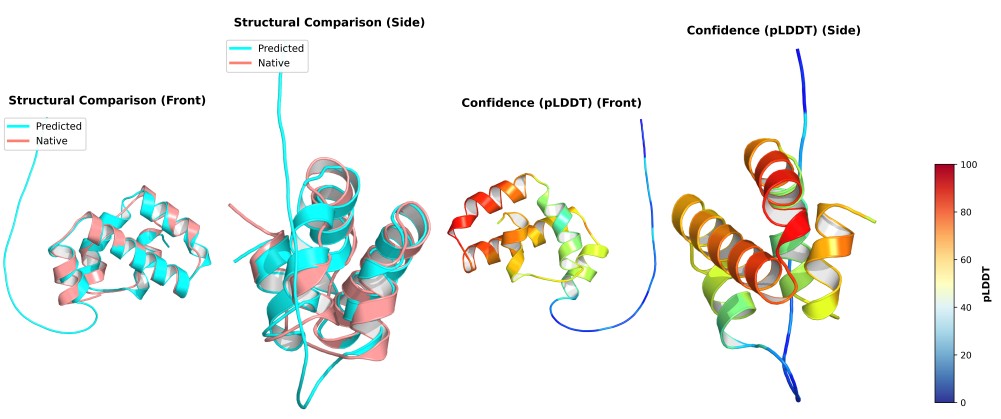

**T1082: pLDDT=66.9 | pTM=0.570 | TM-score=0.815 | CoLD MSA**

Figure 9: T1082 using CoLD-generated MSA: pLDDT=66.9, TM-score=0.815

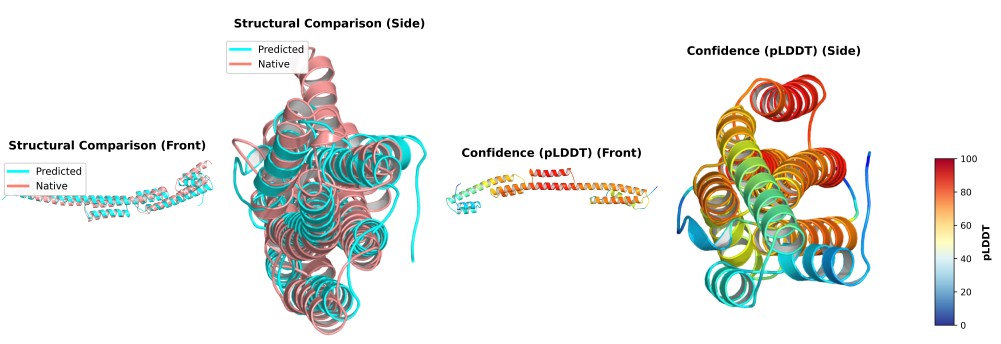

**T1030: pLDDT=78.7 | pTM=0.620 | TM-score=0.743 | MMseqs2 MSA**

Figure 10: T1030 baseline using MMseqs2 MSA: pLDDT=78.7, TM-score=0.743

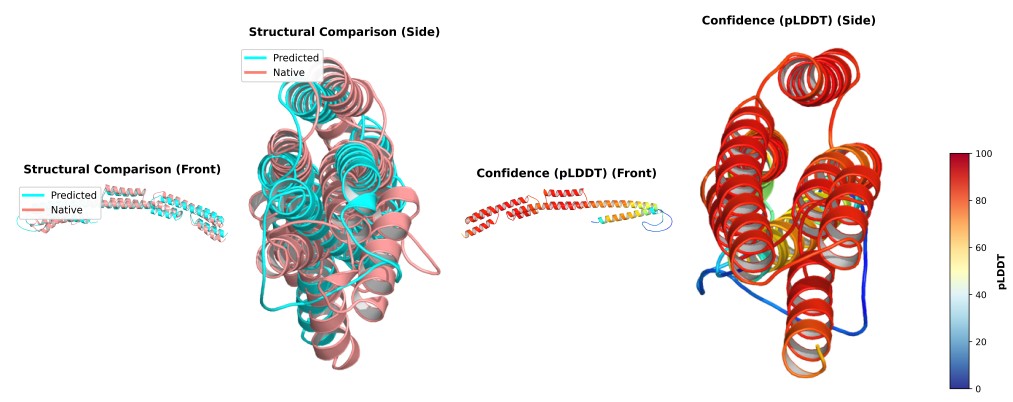

**T1030: pLDDT=85.0 | pTM=0.700 | TM-score=0.646 | CoLD MSA**

Figure 11: T1030 using CoLD-generated MSA: pLDDT=85.0, TM-score=0.646

homology cases. Conversely, T1030 exhibits a trade-off where CoLD achieves higher confidence (pLDDT: 85.0 vs 78.7) but slightly lower structural accuracy (TM-score: 0.646 vs 0.743), suggesting target-specific optimization requirements.

The confidence mapping reveals CoLD's tendency to produce more uniform confidence distributions, avoiding extreme low-confidence regions commonly observed with sparse sampling. This validates CoLD's practical utility for enhancing structure prediction workflows, particularly for challenging targets where traditional methods yield insufficient evolutionary information.

## D  LARGE LANGUAGE MODELS USAGE

Large language models (LLMs) were used only for editorial assistance—e.g., improving grammar, refining phrasing, and clarifying figure captions. The scientific ideas, methodology, model design, data processing, experiments, analyses, and interpretations are entirely the authors' original work. LLMs were not used to generate, modify, or select biological sequences, nor to produce quantitative results or mathematical formulations. All technical content, algorithms, proofs, and empirical findings were developed and validated by the authors.

