# OpenReview forum: "CoLD: A Co-evolutionary Latent Diffusion Model for MSA Generation"
_ICLR.cc/2026/Conference — ICLR 2026 Conference Withdrawn Submission_

### Official Review · Reviewer_yMVg · 2025-10-28

**Soundness:** 2
**Presentation:** 2
**Contribution:** 2
**Rating:** 4
**Confidence:** 4

**Summary:**

This paper introduces CoLD (Co-evolutionary Latent Diffusion), a novel method for generating multiple sequence alignments (MSAs) from a single protein sequence, especially in zero-shot scenarios where no evolutionary homologs are available.

CoLD uses diffusion models in a continuous latent space (via ESM-2 embeddings) to simulate the gradual divergence of homologous sequences.

CoLD models the entire MSA jointly, preserving global co-evolutionary constraints.

By adjusting the diffusion starting timestep, users can control the evolutionary distance of generated sequences.

**Strengths:**

This paper introduces a method that could potentially improve structure prediction for unseen proteins. Authors collected a large MSA dataset and added some architecture ablations.

**Weaknesses:**

Unclear evidence for the link between evolution and diffusion. Method superiority proofs do not look convincing. The number of sequences in the generated MSA is too low (max = 16), making the method useless in practice without scaling. I think that this method have a potential, but the paper needs deep revison of the whole Results section.

**Questions:**

1- You use sequence similarity as a measure of evolution, but something closer to protein evolutionary distance (as in phylogenetics) is necessary.

2- Structure prediction models are usually trained on a datasets with a time cutoff to make benchmarks more fair. Did you use such a cutoff, and did you ensure test sequences and their close homologs were not in the train set?

3- pLDDT is not suitable for this structure prediction evaluation. Please add a separate table for unconditional generation that includes pLDDT, diversity, novelty, and alignment-specific metrics like alignment scores.

4- In Table 1, all metrics look much too low. For example in the original ESMFold paper, the TM score on CASP14 is 67.8 [1] but in this paper it is only 0.35.

5- Low sequence in the alignment (4, 8, 16). Metrics show poor structure prediction quality, likely due to small alignments. You should scale the alignment sizes.

6- There is another work on latent diffusion, that have many connections with your work [2]. Please check it. For example it shows the drawbacks of the cosine scheduler.

7- In Section 4.5, you draw conclusions from only three examples and point out issues in other methods without a comparative evaluation.

8- Please show cases where there are no good natural alignments, and your generated alignments improve structure prediction metrics.

9- It is unclear how much column attention helps; there is no ablation.

10 - Figure 4:  when generating with large t_start, the alignment looks too diverse to be an alignment. Please prove these are real alignments. For example, realign with a standard multiple alignment tool and show that the scores are similar to natural ones for the same query, even if the sequences differ.

11- It is not clear what “ESM guidance” means in Table 2.

12- Quality gets worse when t_start decreases (Table 3). It seems the model is not adapted for “controllable generation” and cannot follow the control well. Maybe the training should include this condition.

13- In Figures 5–7, the proteins look very poor; it is not clear why these examples are shown.

14- Training lengths go up to 1000, but experiments only show up to 370. What happens for longer sequences?

[1]- Language models of protein sequences at the scale of evolution enable accurate structure prediction.
doi: https://doi.org/10.1101/2022.07.20.500902 \
[2]- Diffusion on Language Model Encodings for Protein Sequence Generation.
https://icml.cc/virtual/2025/poster/43588

---

### Official Review · Reviewer_7LiP · 2025-11-03

**Soundness:** 4
**Presentation:** 3
**Contribution:** 3
**Rating:** 8
**Confidence:** 3

**Summary:**

The paper introduces CoLD (Co-evolutionary Latent Diffusion), a novel method for generating Multiple Sequence Alignments (MSAs) to address the challenge of predicting structures for "orphan" proteins with few homologs. Unlike existing autoregressive methods, CoLD frames MSA generation as a conditional diffusion process within the continuous embedding space of a pre-trained protein language model. This approach models evolution as a smooth trajectory on a manifold and captures co-evolution through joint probability distributions. CoLD allows for controllable generation of homologs based on evolutionary distance and is trained with biological constraints for consistency and diversity. Extensive evaluations on CASP 14/15 benchmarks show that CoLD significantly outperforms current methods, dramatically improving confidence metrics for orphan proteins while preserving conserved patterns.

Overall, I think this is a compelling and well-presented paper.

**Strengths:**

1. Well-Defined and Compelling Motivation: The paper effectively addresses a critical bottleneck in protein structure prediction: the poor performance on orphan proteins due to sparse or non-existent Multiple Sequence Alignments (MSAs). The proposed solution—using a diffusion model to generate evolutionarily plausible sequences—is a natural and elegant approach. By framing the problem as one of evolutionary trajectory completion, the authors provide a biologically intuitive and theoretically sound rationale for augmenting MSAs, thereby bridging a fundamental gap in the field.

2.  Novel and Methodologically Sound Approach: The core methodology is both innovative and well-justified. Moving beyond discrete token-based generation, the authors' decision to leverage a continuous latent diffusion model (CoLD) is a significant advancement. This architecture is particularly adept at capturing the global co-evolutionary constraints and complex probability distributions inherent in protein families. The presented framework diagram and the two-stage training paradigm, which incorporates biological constraints, convincingly demonstrate the robustness and logical validity of their design.

3.  Rigorous and Convincing Evaluation: The paper provides strong empirical evidence to support its claims. The extensive evaluation on standard CASP benchmarks and challenging zero-shot scenarios demonstrates a substantial performance improvement over existing methods. The reported 11-point confidence increase for orphan proteins, coupled with the preservation of superior conserved patterns, offers clear and compelling validation of the model's effectiveness and practical utility.

**Weaknesses:**

Need for a More Direct Comparative Analysis and Exploration of Synergy

The study convincingly demonstrates that "CoLD + AlphaFold" surpasses ESMFold on orphan proteins. However, since ESMFold represents a distinct and leading non-MSA-based paradigm for direct structure prediction, a more granular comparative analysis would be highly valuable.

    Fairer Benchmarking: The current comparison could be perceived as somewhat asymmetrical. A fairer evaluation might involve using CoLD to generate MSAs for a standalone, non-AlphaFold prediction model (e.g., RoseTTAFold, OpenFold) and then comparing its performance against ESMFold on the same orphan protein test set. This would more directly isolate and quantify the value added by the generated MSAs themselves, independent of the specific AlphaFold architecture.

    Exploring Synergy: Furthermore, an intriguing and unexplored question is whether CoLD's generated MSAs could also enhance ESMFold's performance. While ESMFold is designed to operate without MSAs, it is plausible that high-quality, generated evolutionary data could still provide a valuable signal. Investigating a potential "CoLD + ESMFold" pipeline would be a fascinating extension, testing whether evolutionary constraints from generative models can complement the internal language model representations of ESMFold.

**Questions:**

None

---

### Official Review · Reviewer_rUuv · 2025-11-04

**Soundness:** 1
**Presentation:** 1
**Contribution:** 1
**Rating:** 0
**Confidence:** 5

**Summary:**

The authors propose CoLD, a method that reformulates MSA generation as conditional diffusion in continuous protein embedding space rather than discrete sequence generation. Using ESM-2 to encode sequences into embeddings, they train an MSA Transformer with axial attention to denoise corrupted MSA embeddings, followed by a decoder to map embeddings back to sequences. The core claim is that this continuous approach better captures co-evolutionary relationships by jointly modeling entire MSA embeddings rather than generating sequences independently. They employ a two-stage training paradigm (decoder pretraining, then progressive diffusion training with multiple biological constraints) and enable controllable evolutionary divergence through variable starting timesteps. Evaluation focuses on downstream structure prediction using AlphaFold2 on CASP14/15 benchmarks and a small zero-shot dataset.

**Strengths:**

The paper addresses an important and well-motivated problem. Generation of high-quality MSAs will make it possible to raise the structure quality predictions fro the orphan proteins and may generally improve speed and quality of structure predictions, including proteins with different states.

**Weaknesses:**

1. **Missing critical ablation of joint vs. independent generation**
The paper's central claim, that joint MSA embedding generation captures co-evolutionary relationships better than independent sequence generation, lacks direct experimental validation. The authors generate a tensor of 100 ESM-2 encodings simultaneously and claim this is superior, but provide no experiment demonstrating this. This is the core hypothesis requiring empirical proof, not just conceptual arguments. Using the same model architecture the authors at the very least should compare generating 100 embeddings jointly (current CoLD approach) against 100 sequences independently (one embedding at a time with identical guidance).


2. **Weak theoretical justification for co-evolution claims.**
The paper asserts that joint MSA embedding generation captures co-evolutionary relationships better than prior methods, but the mechanism is unclear. The model uses standard axial attention on independent position embeddings—it's not evident how this fundamentally differs from existing approaches in preserving co-evolutionary constraints. The theoretical analysis (Appendix A) contains hand-wavy "propositions" without rigorous proofs. Ablating the method thoroughly would help to understand if there is indeed a need in the attention as implemented by the authors.


3. **Noise Addition ≠ Evolutionary Divergence Modeling.**
The claim that "forward diffusion simulates evolutionary divergence" (line 226) is conceptually flawed. Noising embeddings degrades them into undecodable representations, which lose semantic meaning. True evolutionary divergence would produce valid, related protein sequences at each step. The connection between noise level and evolutionary distance is **assumed but never validated** against actual phylogenetic distances or evolutionary metrics.


4. **Unvalidated "Controllable Evolutionary Divergence".**
Table 3 and Figure 4 claim "precise control over evolutionary distance," but the relationship between timestep ratios and actual evolutionary divergence is purely speculative. Sequence similarity percentages (6.80% to 94.13%) measure only sequence identity, not evolutionary distance. Missing are validation against known phylogenetic trees, substitution rate analyses, or proxy metrics like pairwise Levenshtein distance distributions within alignments. The contradictory optimization patterns (pLDDT peaks at 90-100% ratio while TM-score peaks at 40%) further undermine claims of interpretable biological control.


5. **Confounded evaluation without direct MSA quality validation.**
All evaluation relies exclusively on AlphaFold2 structure prediction, making it impossible to isolate MSA quality from AlphaFold2's specific biases. The paper provides no direct MSA quality metrics beyond three cherry-picked conservation examples and never compares generated MSAs against ground-truth natural MSAs from the same protein families. Cannot determine whether improvements stem from genuine evolutionary modeling or from generating artifacts that happen to exploit AlphaFold2's training biases.


6. **Missing statistical significance testing.**
Tables 1 and 2 show performance differences without error bars, confidence intervals, or significance tests. CASP14 differences between CoLD (68.43), MSAGPT (65.42), and PLAME (64.14) pLDDT are 3-5 points and could be within experimental noise. The zero-shot dataset contains only 30 proteins—insufficient for drawing strong conclusions about the claimed 11-point improvement. No hypothesis testing validates whether observed improvements are statistically meaningful or justify the added model complexity.


7. **Unstable ablation results indicate poor component integration.**
Table 2 reveals that removing components sometimes **improves** specific metrics: w/o ESM Guidance achieves highest CASP15 pLDDT (70.12 vs 69.79 full model), and w/o Profile Loss achieves best CASP14 LDDT (64.21 vs 63.48). This suggests components work at cross-purposes, hyperparameters were overfit to the full configuration, or optimization was insufficient. The diversity loss ablation shows competitive performance on most metrics, questioning whether all "biological constraints" are beneficial.


8. **Inadequate conservation analysis.**
Figure 5 presents only three cherry-picked examples with high Pearson correlations (0.938-0.994) but no systematic evaluation across the full test set. Critical information is missing: at what t_start were these generated? How stable are conservation patterns across different generation parameters? Do these represent typical performance or best-case scenarios? The claim of "up to 0.994 correlation" in the abstract lacks context about the distribution across all targets.


9. **Missing diversity and phylogenetic validation.**
Fundamental biological validation is absent. The paper provides no diversity analysis showing whether generated sequences form plausible evolutionary families, no phylogenetic validation demonstrating that generated MSAs recapitulate known evolutionary relationships, and no analysis of whether amino acid distributions, gap patterns, or other statistical properties match natural MSAs. The conceptual leap from "denoising to generate many proteins" to "generating a biologically coherent MSA" remains unvalidated.


10. **Unjustified progressive training complexity.**
Progressive training is the most critical component (Table 2: removes it drops pLDDT from 68.43 to 62.85), yet no ablation tests whether simpler schedules work equally well. The claim that this "mirrors biological hierarchy" (line 262) is fabricated motivation—this is standard curriculum learning with no biological analog. The model appears optimized specifically with this curriculum and fails without it, suggesting dependence on training procedure rather than principled biological design.


11. **Ad-hoc gap insertion mechanism lacks justification.**
The "learned threshold mechanisms based on embedding magnitudes" for gap insertion (line 222) appears unprincipled. ESM-2 already includes a "-" token in its vocabulary—why not use it directly? No ablation compares direct gap token usage against the current approach. The mechanism may not generalize to different embedding spaces or protein families and seems like a workaround rather than a principled solution.


12. **Missing diffusion schedule ablations.**
No experiment validates that the cosine schedule outperforms linear or other schedules. The biological motivation (line 233) that "early timesteps preserve fine-grained evolutionary relationships while later timesteps approach pure noise" applies equally to any monotonic schedule and provides no specific justification for choosing cosine. This standard hyperparameter choice should have been validated experimentally.


13. **Unexplained baseline anomalies.**
AF2+MSA shows surprisingly low CASP15 performance (41.25 pLDDT)—lower than single-sequence methods—with no explanation. This anomaly raises questions about the evaluation protocol or baseline MSA quality that could affect interpretation of all results. The experimental setup may have issues that systematically disadvantage certain baselines. Does it happen because the MSAs used in the experiment are not deep enough, or because their properties differ much from the natural MSAs, or for some other reason?


14. **Incomplete related work coverage.**
Recent relevant methods are missing from comparison and discussion, including DiMA (arxiv.org/abs/2403.03726) and ProDiT (biorxiv.org/content/10.1101/2025.09.03.672144v2) among others, creating incomplete context for evaluating CoLD's novelty. The paper doesn't position itself adequately within the rapidly evolving landscape of diffusion-based protein modeling. Also, both PLAME and CoLD operate in ESM embedding space, yet PLAME achieves competitive zero-shot results (50.70 vs 62.03 pLDDT) with apparently less complexity. The paper doesn't adequately explain why diffusion fundamentally outperforms PLAME's approach or whether the 11-point gap justifies the added architectural and computational overhead. Claims about "continuous manifold modeling" advantages remain vague without concrete mechanistic explanations.


15. **Theoretical analysis lacks rigor and sounds LLM-generated.**
Appendix A presents "propositions" and "claims" without formal proofs. Proposition A.1 provides no actual proof that diffusion preserves co-evolutionary correlations (Equation 12 merely shows noise affects correlations). Analysis A.2 offers a hand-wavy "convergence argument" without formal analysis. Claim A.3's information-theoretic optimality is stated but not proven, and Equation 15's inequality has no derivation or empirical validation. These theoretical contributions cannot be evaluated without proper mathematical development. The same applies to the flawed statements at lines 226, 233, 262, and so on.


16. **Presentation issues indicate rushed preparation.**
Lines 188 and 192 contain repeated sentences. Figure 3 (protein-wise optimization plots) appears without discussion in the main text. Minimal detail in appendices about case study selection and protocols. These issues, combined with missing ablations on core claims and insufficient integration between main text and supplementary material, suggest incomplete preparation.

**Questions:**

* How do generated MSAs compare statistically to natural MSAs from the same protein families? What would happen if we run CoLD generation with guidance on different proteins from the same families? How diverse would be such MSAs?
* How were the 30 zero-shot proteins selected and verified as absent from training data?
* What explains the anomalously low AF2+MSA baseline performance on CASP14/15?

---

### Official Review · Reviewer_px54 · 2025-11-04

**Soundness:** 1
**Presentation:** 1
**Contribution:** 1
**Rating:** 0
**Confidence:** 4

**Summary:**

This paper presents CoLD, a novel generative model for Multiple Sequence Alignments (MSAs). The work targets the critical challenge of structure prediction for "orphan proteins," which lack the deep homologous sequences required by state-of-the-art models like AlphaFold2. The authors reformulate MSA generation as a conditional diffusion process operating in the continuous latent space of a pretrained protein language model (ESM-2). Unlike discrete autoregressive models, CoLD models the joint distribution over the entire MSA embedding tensor simultaneously. The authors claim this approach better captures the continuous nature of evolution and global co-evolutionary constraints. The method allows for controllable generation of evolutionary diversity and reportedly demonstrates significant (11+ point) improvements in confidence metrics for zero-shot (orphan) benchmarks.

**Strengths:**

The work addresses a well-known and highly significant bottleneck in computational structural biology. The inability to accurately predict structures for "orphan proteins" is a major limitation of current SOTA methods, and a robust solution would be a major contribution to the field.

The conceptual shift from discrete, autoregressive generation to a joint, continuous diffusion process is highly novel and well-motivated. The hypothesis that co-evolution is a joint property best captured by modeling the entire alignment tensor simultaneously (rather than sequence-by-sequence) is biologically plausible and, if successful, addresses a key limitation of existing methods.

The reported results, particularly the 11-point pLDDT improvement in the zero-shot scenario, are substantial. The model's ability to achieve high correlation (up to 0.994) with natural conservation patterns is also impressive.

**Weaknesses:**

Despite the promising technical idea, this paper suffers from severe and disqualifying flaws in its literature review and citation practices. The citations for numerous key baselines (the very state-of-the-art methods against which this work is benchmarked) are demonstrably incorrect. This negligence undermines the authors' claims of novelty, the validity of their comparative evaluation, and the scholarly integrity of the entire manuscript.

This is not a matter of minor formatting typos; these are fundamental errors of attribution for the paper's primary competitors.

- PLAME: The paper cites Cao et al., Nature Communications, 2025. This citation is incorrect in both its author list and its publication venue.

- MSAGPT: The paper cites Chen et al., Nature Machine Intelligence, 2024. This is also incorrect. The well-known MSAGPT paper has a different author list and publication history.

- EvoGen: The paper cites Zhang et al., Nature Machine Intelligence, 2022. This is a critical error. The widely referenced EvoGen paper (from J. Chem. Theory Comput.) has a completely different author list (e.g., Zhang, Liu, Chen, Gao, et al.).

- MSA-Generator: The citation to Zhang et al., arXiv, 2023 appears to be misattributed.

- TaxDiff: The citation to Zhang et al., arXiv, 2024 follows the same pattern of incorrect attribution.

The systematic nature of these errors, spanning nearly all major generative baselines, is deeply concerning. This pattern is strongly suggestive of a literature review that was automatically generated (e.g., by an LLM) and published without basic scholarly verification. It gives the impression that the authors have "hallucinated" a field of competitors.

Given the severity of these scholarly lapses, the paper is not in a suitable state for publication at ICLR. The promising technical idea of CoLD is completely overshadowed by this failure in basic scholarship. The authors are strongly advised to perform a complete and manual rewrite of their related work and re-verify their entire experimental comparison. Following such a thorough revision, the authors might consider submitting their work to a journal more specialized in computational biology, rather than resubmitting to ICLR.

**Questions:**

N/A

**Details Of Ethics Concerns:**

Mention in the weakness section

---

### Official Review · Reviewer_gBRT · 2025-11-11

**Soundness:** 1
**Presentation:** 2
**Contribution:** 2
**Rating:** 2
**Confidence:** 4

**Summary:**

This paper proposes a novel method for simulating an MSA given a query sequence. The method works by (1) embedding the query with ESM, (2) creating N copies of this embedding and perturbing each independently with isotropic Gaussian noise, (3) denoising these N copies using a (reverse) diffusion process, and (4) decoding the denoised copies back to sequence space with a (separately trained) decoder. The major challenge is training the denoising model, which requires a progressive training scheme involving different regularizers. The trained model improves AlphaFold performance on structure prediction benchmarks (CASP14/15 & a zero-shot setting).

**Strengths:**

- I am not familiar with this specific area, but the idea of using ESM's latent space to generate homologs (and thereby MSAs) is cool.
- Improved performance on benchmarks.

**Weaknesses:**

Overall, the technical quality of the work (the arguments, maths, clarity) is, in my opinion, quite weak.

In their discussion of "co-evolution", I don't understand why the authors state that "Existing generative approaches that model sequences independently fundamentally fail to capture these global correlations". The whole point of deep models such as ESM is that they can model dependencies between sites. The typical complaint about deep models like ESM is that they assume the sequences in the MSA are independent, i.e. P(s_1, ..., s_n) = \prod_i P(s_i). This has nothing to do with co-evolution between sites.

My impression is that there's no need to model the whole MSA in CoLD. If you just embed the query, corrupt it, and denoise it towards some reasonable sequence (e.g. some sequence from the MSA, or even the query sequence), then doing this n times *independently* would also produce a reasonable MSA. I understand that the regularizers in the objective of CoLD involve matching profiles, but this doesn't conceptually require joint modeling of the sequences in the MSA. It might require sampling several times from the model to get an *estimate* of the loss, but this is a different matter. In all, none of the arguments provided in the paper supports the idea that modeling P(Z | z_q) jointly is important and a major contribution of the work. The work seems to convey a conceptual flaw of what "co-evolution" is and its relationship to MSAs.

The loss function used and training scheme are quite finicky. The fact that a "progressive training scheme" is needed for good performance (it is the *most important* component, as revealed by the ablations in Table 2) as well as the multiple regularization terms in the loss makes me skeptical of the robustness of the method. I can imagine the method being worked on by progressively adding more components to it until it became SoTA. Then a retrospective ablation was performed for the submission. In fact, I wonder how much of the model's performance comes from overfitting. I might be completely wrong - I hope some other reviewer familiar with the benchmarks can clarify.

When math is used, it is used in an imprecise and sloppy way:
- Proposition A.1 => Prove equation 12. Why does this equality hold?
- "A.2 MANIFOLD CONVERGENCE PROPERTIES" => I'm not sure why this paper talks about manifolds at all. If anything ESM induces a probability measure on \cup_L R^{L \times d} by pushing forward the distribution of natural proteins into \cup_L R^{L \times d}. Whether this (or a family's submeasure) is supported on some sub-manifold M_evo seems irrelevant. The reverse diffusion process by definition maps Z_t onto Z_0, so I don't understand what section A.2 is about.
- "A.3 INFORMATION-THEORETIC OPTIMALITY" => I don't understand the point of this section. In equation (15), P_{natural}, P_{joint}, and P_{independent} are not defined. Please define this first. My understanding is that section A.3 is simply saying that the model class of joint distributions is larger than the class of independent distributions, thus a lower KL divergence can be achieved to the true distribution, which is a trivial statement. What is the intention of this section? In this sense, I can't see how equation (14) used to prove (15). Please formalize your result and prove equation (15). Otherwise, remove this result.
- "A.4 EVOLUTIONARY DISTANCE CONTROL" => In equation (16), what is the expectation over? The generated sequence? What is d_evo? Is d_evo the euclidean distance in ESM latent space R^{L \times d}? The first definition of d_evo happens in line 150, and it's not even a proper definition -- it just says it's the "evolutionary distance". What is "evolutionary distance"?
- Generally, referring to Appendix A as "The theoretical optimality of this approach is established in Appendix A." is rather amusing given how sloppy the theoretical results presented are. In my opinion, the work is improved by completely removing Appendix A.
- Why do you call the condition in line 149 "local smoothness"? This seems more like an "approximate isometry" condition. Smoothness is usually related to gradient conditions. Is this some standard term in this literature?
- Line 271 "consitency loss ensures" => it does not "ensure", if anything it *encourages*.
- Generally, there is sxcessive use of vague phrases such as "evolutionary geometry", "continuous manifold diffusion". It is hard to distinguish what is a precise mathematical term or result, from some vague high-level intuition.

My impression is that the idea of perturbing the ESM embeddings to generate syntetic MSAs is good, but the execution in this work is technically sloppy and has so many moving parts, that I cannot currently recomment this work for acceptance.

**Questions:**

In addition to the points brought up in the "weaknesses" section:
- As I mentioned previously, modeling P(Z|z_q) jointly is not needed to capture co-evolution between sites. Single-sequence models can also capture this, as "co-evolution" is about intra-sequence correlations, not inter-sequence correlations. Am I missing something?
- In Table 2, what is the "ESM Guidance" ablation?
- Figure 3: This looks quite noisy, e.g. green line for TM-score plot. Why is this? Did you sample a single MSA from each t_start?
- I am confused by the notion of "Ratio" in Table 3 and Figure 4: In table 3, low ratio means low similarity, while in Figure 4, low "r" means high similarity.
- Table 3: Is this the average test perf over all proteins?
- Figure 5: What value of "Ratio" was used to generate these plots? Generally, how was the ratio for each family chosen during training and testing?
- "This conservation preservation stems from joint MSA embedding modeling" => Again, I don't really see how the "joint" part matters. You can take a simple model like the Le and Gascuel independent sites model and mutate a starting protein many times independently to simulate an MSA, and it will retain conservations patterns by virtue of modeling site rate variation.
- "The intermediate regime (r=0.4-0.7) [...] corresponds to the optimal TM-Score performance observed in Table 3." => This is not supported by the table: the top TM values in Table 3 are in fact obtained for the most extreme ratios of 10%-40%, not the intermediate regime.
- Why was no code provided?

---

### Note · Authors · 2025-12-04

**Comment:**

We would like to thank the reviewers for taking the time to read our submission and provide detailed feedback. After carefully considering the comments, we feel that several parts of the manuscript, including the theoretical explanation, the citation accuracy, and some aspects of the experimental analysis, require more substantial revision than what can be reasonably completed during the rebuttal period.

To ensure that the work is presented with the clarity and rigor that it deserves, we have decided to withdraw the current version. We will revise the manuscript thoroughly and incorporate the reviewers’ suggestions as we prepare a more polished version for a future submission.

We are grateful for the reviewers’ insights and appreciate the time they devoted to our work.

**Withdrawal Confirmation:**

I have read and agree with the venue's withdrawal policy on behalf of myself and my co-authors.